# The Need for Novel Asexual Blood-Stage Malaria Vaccine Candidates for *Plasmodium falciparum*

**DOI:** 10.3390/biom14010100

**Published:** 2024-01-12

**Authors:** Eizo Takashima, Hitoshi Otsuki, Masayuki Morita, Daisuke Ito, Hikaru Nagaoka, Takaaki Yuguchi, Ifra Hassan, Takafumi Tsuboi

**Affiliations:** 1Division of Malaria Research, Proteo-Science Center, Ehime University, Matsuyama 790-8577, Japan; morita.masayuki.ls@ehime-u.ac.jp (M.M.); nhikvip@tmd.ac.jp (H.N.); ilyiukie@gmail.com (T.Y.); ifra.razaq3627@gmail.com (I.H.); 2Division of Medical Zoology, Department of Microbiology and Immunology, Faculty of Medicine, Tottori University, Yonago 683-8503, Japan; otsuki@tottori-u.ac.jp (H.O.); dito@tottori-u.ac.jp (D.I.); 3Division of Cell-Free Sciences, Proteo-Science Center, Ehime University, Matsuyama 790-8577, Japan

**Keywords:** antigen discovery, asexual blood stage, malaria, vaccine, *Plasmodium falciparum*

## Abstract

Extensive control efforts have significantly reduced malaria cases and deaths over the past two decades, but in recent years, coupled with the COVID-19 pandemic, success has stalled. The WHO has urged the implementation of a number of interventions, including vaccines. The modestly effective RTS,S/AS01 pre-erythrocytic vaccine has been recommended by the WHO for use in sub-Saharan Africa against *Plasmodium falciparum* in children residing in moderate to high malaria transmission regions. A second pre-erythrocytic vaccine, R21/Matrix-M, was also recommended by the WHO on 3 October 2023. However, the paucity and limitations of pre-erythrocytic vaccines highlight the need for asexual blood-stage malaria vaccines that prevent disease caused by blood-stage parasites. Few asexual blood-stage vaccine candidates have reached phase 2 clinical development, and the challenges in terms of their efficacy include antigen polymorphisms and low immunogenicity in humans. This review summarizes the history and progress of asexual blood-stage malaria vaccine development, highlighting the need for novel candidate vaccine antigens/molecules.

## 1. Introduction

Malaria, a disease of global health priority, is caused by infection, with the *Plasmodium* parasite transmitted by anopheline mosquitoes. Global malaria deaths decreased steadily from 864,000 in 2000 to 576,000 in 2019; however, a new increase in deaths to an estimated 608,000 in 2022 is thought to be caused by disruptions to malaria control efforts during the COVID-19 pandemic [1]. These recent events emphasize the need to develop highly effective malaria vaccines to control and eventually eradicate the disease.

Malaria vaccines can be classified into three groups, each of which focuses on the specific developmental stages of the parasite: the pre-erythrocytic (sporozoite and liver stages), asexual blood (merozoite, trophozoite, schizont, and parasite-infected erythrocyte), and sexual stages within the bloodstream and the mosquito (gametocyte, gamete, and zygote/ookinete) (Figure 1).

The Malaria Vaccine Technology Roadmap has been updated for the development of new malaria vaccines by 2030. The renewed roadmap calls for developing vaccines with at least 75% protective efficacy against clinical malaria, targeting both *Plasmodium falciparum* and *Plasmodium vivax*, and vaccines that reduce the transmission of the parasite [2,3]. One leading malaria vaccine, RTS,S/AS01, is a pre-erythrocytic vaccine that uses the circumsporozoite protein (CSP) of *P. falciparum*, which is expressed during the sporozoite stage. The RTS,S antigen consists of the C-terminal region and central repeats of CSP and is expressed with fused (20%) and free (80%) hepatitis B surface antigen (HBsAg), which self-assembles into virus-like particles [4,5] and is formulated with the AS01 adjuvant. Phase 3 trials showed moderate efficacy against clinical malaria [6] with short duration [7]. Since 2019, large pilot implementation programs have been conducted in Ghana, Kenya, and Malawi [8]. A significant reduction in severe malaria cases by approximately 30% in the first 2 years [9] was noted. On 6 October 2021, the World Health Organization (WHO) recommended the broad use of this vaccine among children over 5 months old living in moderate to high falciparum malaria transmission regions [10].

Currently, the second-generation malaria vaccine candidates in clinical development include the most advanced R21/Matrix-M [11,12]. R21 is also a pre-erythrocytic malaria vaccine antigen and shares similarities with RTS,S, including the fusion of HBsAg to the C-terminal domain and central repeats of CSP. However, unlike RTS,S, R21 consists solely of the fusion protein with HBsAg, resulting in greater CSP coverage within the virus-like particles [13,14]. After preclinical studies of R21 formulated with multiple adjuvants, Matrix-M (R21/Matrix-M) was selected for clinical development because of its higher immunogenicity. In a phase 2b trial involving 5- to 17-month-old Burkinabe children, R21/Matrix-M had an efficacy of 77% against clinical malaria [13]. Based on these trial results, the vaccine recently obtained regulatory clearance in Ghana, Nigeria, and Burkina Faso (https://www.gavi.org/vaccineswork/five-things-you-need-know-about-new-r21-malaria-vaccine; https://allafrica.com/stories/202304190024.html; https://www.ox.ac.uk/news/2023-07-24-oxford-r21matrix-m-malaria-vaccine-receives-regulatory-clearance-use-burkina-faso; accessed on 1 October 2023) and was recommended by the WHO on 3 October 2023 [1].

RTS,S and R21 are both classified as pre-erythrocytic vaccines and this distinction is important; in a study conducted by White et al. [7], the level of anti-CSP antibodies provided by the RTS,S/AS01 vaccine helped inform its efficacy profile and estimate the duration of protection. In low-transmission areas, efficacy against clinical malaria declines because of a reduction in anti-CSP antibody titers. In high-transmission areas, efficacy against clinical malaria decreases much faster in vaccinated individuals because of RTS,S blocking sporozoite invasion, which results in lowered blood-stage immunity. Hence, White et al. [7] demonstrated that pre-erythrocytic anti-infection vaccines for malaria have limitations and highlighted the significance of developing vaccines that target asexual blood stages, where the associated manifestations of the disease are predominantly attributed. Moreover, it is theoretically possible to enhance the protective immunity provided by blood-stage vaccines through natural infection with the parasite [15].

The value of blood-stage vaccines cannot be overemphasized. In addition to the renewed Malaria Vaccine Technology Roadmap [2], a recent WHO publication, “Malaria vaccines: preferred product characteristics and clinical development considerations” [16], stated two priorities. Strategic Goal 1 called for “malaria vaccines that prevent human blood-stage infection of the individual level”, and Goal 2 described “malaria vaccines that reduce morbidity and mortality in individuals of risk in malaria-endemic areas”. Another recent publication, the “WHO Guidelines for Malaria” [17], also emphasized the need for asexual blood-stage vaccine development for malaria eradication. In this review, we summarize the history and progress of the development of asexual blood-stage malaria vaccines and stress the importance of discovering novel blood-stage vaccine candidate molecules to help address these roadmap goals [15].

## 2. Complex Processes of the Asexual Blood-Stage Malaria Lifecycle and the Molecules Involved in These Steps

The clinical manifestations of malaria are caused by the highly regulated amplification cycles of the merozoite invasion of erythrocytes, the development of the trophozoite and schizont stages, and the subsequent egress of ~32 mature daughter merozoites. The egress and invasion processes result in critical parasite proteins being exposed to host immune responses [18,19,20] (Figure 2). This protein repertoire is considered to represent asexual blood-stage malaria vaccine candidates. The catalog comprises invasion-related proteins found on the merozoite surface or within merozoite secretory organelles, such as micronemes, rhoptries, and dense granules. Merozoite surface proteins (MSPs) can be anchored by glycosylphosphatidylinositol (GPI), integrated into parasite membranes, or associated with membrane proteins. During schizogony, the secretory organellar proteins of the merozoite are initially retained in the micronemes, rhoptries, or dense granules. Following this, some microneme and rhoptry proteins are relocated to the merozoite surface after egress [21]. Once released into the bloodstream, the merozoite invasion cascade, which includes pre-invasion, internalization, and echinocytosis, is completed within a few minutes (as summarized in [22]).

The pre-invasion step, which involves merozoite attachment and reorientation, is the first contact between merozoites and erythrocytes and is mediated by molecular interactions that are not fully understood. The dynamic interplay between the merozoite and erythrocyte during the pre-invasion step results in erythrocyte deformation driven by a parasite actin-myosin motor [22,23]. Then, the merozoite reorients the apical end toward the erythrocyte membrane through erythrocyte-binding antigens (EBAs) and reticulocyte-binding-like homologs (Rhs) [23]. The merozoite anchors its apical via the interaction of Rh5 with an erythrocyte surface protein, basigin, resulting in irreversible merozoite attachment to the erythrocyte. A tight junction is formed between the parasite apical membrane antigen (AMA) 1, and the rhoptry neck protein (RON) complex, RON2 and RON4. The merozoite invades the erythrocyte using the parasite actin-myosin motor, creating a parasitophorous vacuole (PV) membrane that surrounds the parasite [24]. After the invasion, membrane fusion occurs at the posterior end of the merozoite to enclose it within the PV and erythrocyte plasma membranes. Echinocytosis occurs once the parasite has started invasion into the erythrocyte, which leads to erythrocyte shrinkage and the formation of spiky protrusions on the surface [22] (Figure 2). Over the next 48 h, the parasite develops into the schizont stage of up to 32 merozoites (for *P. falciparum*) before egressing to invade new erythrocytes.

EBAs and Rhs bind specific erythrocyte receptors, including glycophorins A, B, and C, and complement receptor 1 (CR1). In *P. falciparum*, there are redundant functions in the individual members of these protein families that need to be considered in the process of selecting vaccine candidates [25]. For example, the 175-kDa EBA (EBA175) binds to glycophorin A on the erythrocyte surface, whereas Rh4 binds to CR1 [26]. After parasite reorientation, Rh5 forms a complex with Rh5-interacting protein (Ripr) [27] and cysteine-rich protective antigen (CyRPA) [28]. Rh5 binds to basigin on the erythrocyte surface, which is crucial for erythrocyte invasion by the merozoite [29]. Recently, *Plasmodium* thrombospondin-related apical merozoite protein (PTRAMP) and cysteine-rich small secreted protein (CSS) in *P. falciparum* were identified as the interacting proteins of the Rh5 complex. The PTRAMP/CSS/Ripr/CyRPA/Rh5 complex plays a significant role in anchoring the contact between merozoite and erythrocyte membranes and is, therefore, essential for merozoite invasion [30,31].

## 3. Discovery of Asexual Blood-Stage Malaria Vaccine Candidate Molecules

The discoveries of asexual blood-stage malaria vaccine candidates before the whole genome sequence era are summarized in a review paper by Good and Miller [32]. Since this information is valuable for considering the future direction of malaria vaccine research, we extracted the essence of their descriptions in the following paragraphs.

Historically, the immunization of *Rhesus* monkeys with *Plasmodium knowlesi* parasite antigens emulsified in Freund’s complete adjuvant (FCA) conferred protection [33,34]. Immune serum from infected monkeys could passively transfer immunity [35]. With the establishment of a continuous in vitro culture for *P. falciparum* [36], monkeys could be immunized with this human parasite [37,38]. In addition, the injection of immunoglobulin purified from adult human serum with naturally acquired immunity to malaria was successful in treating infected non-immune children in Gambia [39] and Thailand [40]. However, because of the limitations of whole parasites as vaccine antigens and a suitable human-compatible adjuvant, researchers have turned to cloning the genes of malaria antigens [32].

Malaria blood-stage antigens were initially cloned in 1983 from *P. falciparum* and *P. knowlesi* malaria parasites using λgt11 expression libraries [41,42]. For *P. falciparum*, the leading subunit vaccine antigens, including MSP1 [43], MSP2 [44], MSP3 [45], and AMA1 [46], were identified by screening the *P. falciparum* expression libraries using either antibodies from malaria-immune Papua New Guinean adults or protective mouse monoclonal antibodies. Similarly, serine repeat antigen 5 (SERA5) [47] and glutamate-rich protein (GLURP) [48,49] were independently identified by screening *P. falciparum* expression libraries with malaria-immune human antibodies from African countries.

A “functional approach” based on the erythrocyte-binding phenotype of the parasite proteins (ligands) was also initiated. Two pioneering works were reported in 1990 on the discovery of parasite ligands, *P. knowlesi* and *P. vivax* Duffy binding proteins (DBP) [50,51], and *P. falciparum* 175 kDa erythrocyte binding antigen (EBA175) [52].

The discovery of the DBP gene involved the identification of soluble parasite proteins found in both *P. knowlesi* and *P. vivax* culture supernatants that specifically bind to Duffy-positive human erythrocytes. The 135 kDa *P. knowlesi* Duffy antigen-binding protein was purified using Duffy-positive human erythrocytes. Briefly, anti-135 kDa antibodies were affinity purified from the immune serum of a rhesus monkey using strips of the nitrocellulose membrane from the 135 kDa region of the gel. A λgt11 expression library derived from *P. knowlesi* asexual blood-stage cDNA was screened with these specific antibodies, leading to the cloning of the PkDBP gene [50]. By using this information, a gene encoding the DBP of *P. vivax* (PvDBP) was also cloned [51].

The 175 kDa *P. falciparum* antigen, EBA175, is a soluble parasite protein that accumulates in the supernatants of schizont cultures [53]. In order to identify the gene for EBA175, EBA175 antigens were affinity purified from the culture supernatants of *P. falciparum* using intact erythrocytes. Monospecific antibodies against EBA175 were then isolated from the sera of *Aotus* monkeys immunized against *P. falciparum* parasites using affinity purification against EBA175 absorbed to a nitrocellulose membrane [54]. Affinity-purified monospecific antibodies against EBA175 were used to screen a *P. falciparum* λgt11 expression library, and an *eba175* gene was cloned [52]. This gene was classified as a member of the erythrocyte binding-like (EBL) protein family, along with PkDBP and PvDBP, based on its primary structure and amino acid homology, all of which have conserved 5′ and 3′ cysteine-rich regions [55]. Finally, several EBA175 paralogs have been identified from the *P. falciparum* genome [56,57,58].

The discovery of *P. vivax* reticulocyte-binding proteins 1 and 2 (PvRBP1 and PvRBP2) was made using an approach similar to that used for identifying EBLs [59]. The orthologous Py235 gene family of these proteins was identified in the rodent malaria parasite *Plasmodium yoelii* [60,61]. In the case of *P. falciparum*, a search of the then incomplete genome data led to the identification of an ortholog of PvRBP1, which was named normocyte binding protein 1 (PfNBP1) [62]. This protein was subsequently categorized as a member of the reticulocyte binding-like (RBL) family of invasion-related proteins [63]. A similar bioinformatic approach was employed to identify and characterize four RBL family members in *P. falciparum* (PfRh), namely PfRh1/PfNBP1, PfRh2a, PfRh2b, and PfRh4 [62,64,65,66].

A number of these blood-stage antigens discovered in the pre-genome era have continued to progress to vaccine clinical trials.

After the availability of parasite genome databases, an additional PfRh member, Rh5, was identified [67,68,69]. Rh5 forms a complex with CyRPA [28] and Ripr [27], as described above, and is abbreviated as the RCR protein complex. Each RCR complex member is highly conserved, essential for invasion, and antibody-susceptible [70,71]. Despite its poor natural immunogenicity, Rh5 remains a blood-stage vaccine antigen candidate because of its indispensable role and highly conserved sequences in field isolates [20,72,73,74].

Apart from the merozoite proteins, a search of the malaria parasite genome was conducted using an α-helical coiled-coil structure, which led to the identification of P27A protein as a fragment of trophozoite exported protein 1 (TEX1), with the potential to serve as a target for blood-stage vaccines [75]. Similarly, other nonmerozoite proteins, *P. falciparum* schizont egress antigen-1 (PfSEA-1) and *P. falciparum* glutamic acid-rich protein (PfGARP), have been identified as blood-stage vaccine candidates from the library screenings of proteins that are only reactive to the sera from malaria-protected children [76,77]. These studies indicated that not only merozoite-stage proteins but also trophozoite antigens could be developed as blood-stage vaccines [78]. The clinical development of a number of antigens is described in the following section.

## 4. Current Status of Leading Asexual Blood-Stage Malaria Vaccine Candidates

The road to asexual blood-stage vaccine development has been arduous, with most clinical trials showing limited efficacy, primarily due to extensive polymorphisms across parasite isolates, the involvement of functionally redundant merozoite invasion pathways, and, in some cases, safety concerns [79]. Only a few vaccine candidates are currently undergoing clinical trials for further evaluation, and their clinical development outcomes are summarized in this section and Table 1. The status of protein structural analyses is incorporated into Table 1 because the latest approaches to improving vaccine antigen design are based on structural information.

### 4.1. AMA1

AMA1 from *P. falciparum* has shown promise as a potential blood-stage vaccine candidate, despite being highly polymorphic. Anti-AMA1 antibodies have strongly inhibited merozoite invasion in vitro, and the knockdown of the *ama1* gene led to a loss of merozoite invasion [100,101]. Several AMA1 vaccine clinical trials have been conducted up to phase 2b; however, the results have been mixed. One phase 2b trial based on AMA1-C1 (a combination of 3D7- and FVO-types of AMA1 adjuvanted with Alhydrogel) in Malian children showed no protection against clinical malaria [102] or strain-specific protection [103]. Another trial using AMA1-3D7 formulated with an AS02A adjuvant (FMP2.1/AS02A) showed no reduction in malaria episodes, and only children infected with the vaccine strain sequence had a significant decrease in clinical episodes [80,81].

In order to address concerns about antigen polymorphisms, three AMA1 ‘Diversity Covering’ (DiCo) proteins, which account for major AMA1 polymorphisms, have been synthesized [104]. In rabbits and monkeys, the three AMA1 DiCo protein mixtures successfully induced cross-reactive antibodies that had in vitro parasite growth inhibition assay (GIA) activity against several *P. falciparum* strains [104,105]. The broadened antibody response is attributed to increased levels of cross-reactive antibodies [106]. However, in Burkina Faso clinical trials, the levels of anti-AMA antibodies against the natural AMA1 variants were only slightly increased despite the significant antibody increase observed for anti-PfAMA1 DiCo antibodies. Moreover, the presence of antibodies against other malaria antigens could have countered the GIA activity of anti-AMA1 [88], resulting in limited efficacy. The utility of a single antigen, AMA1, as a vaccine formulation remains a topic of debate [89,90].

Administering AMA1 with RON2 has the potential to improve the AMA1-based vaccines [107], based on RON2 mediating the binding and tight junction formation of AMA1 protein to establish the attachment of the parasite to red blood cells. In a *P. falciparum Aotus* model, an AMA1 + RON2 combination vaccine induced higher levels of antibodies against AMA1 than AMA1 alone [108]. The binary complex vaccine increased neutralizing antibodies targeting the AMA1-RON2 interaction, and this was significantly correlated with vaccine efficacy [108]. In order to prepare for future clinical trials, the identity and integrity of the candidate antigens were characterized, with the results indicating that the complex was stable for 72 h at 4 °C [109]. Recently, growth-inhibitory epitopes outside the RON2 binding site have been reported using the AMA1-RON2 fusion protein as an immunogen [110]. However, further progress in the development of the AMA1-RON2 complex vaccine has not been reported.

### 4.2. MSP1

Antibodies that target proteins on the surface of merozoites can prevent invasion or inhibit parasite growth in vitro [111,112]. One well-studied antigen, MSP1, is a high-molecular-weight protein that mediates initial attachment by binding to heparin-like proteoglycans or Band 3 [113,114]. Before merozoite egresses, MSP1 is cleaved into several smaller fragments. In the *Aotus* monkey, protective immunity is achieved against MSP1_19_ [115]. Several vaccine formulations of MSP1 have been tested in malaria-naïve subjects using controlled human malaria infections (CHMI). A phase 2b trial using a single MSP1 antigen-based vaccine (MSP1_42_-3D7/AS02) on healthy Kenyan children aged 12–47 months showed no significant clinical efficacy [83]. Similarly, a phase 2a trial in UK adults was conducted using a combination blood-stage vaccine (MSP1, AMA1, a combination of MSP1 + AMA1, or MSP1 + thrombospondin-related adhesive protein (TRAP)) primed with replication-deficient chimpanzee adenovirus serotype 63 (ChAd63) vectored vaccine and boosted with attenuated orthopoxviral-modified vaccinia virus Ankara (MVA) vectored vaccine. The vaccine was safe but did not show blood-stage efficacy after sporozoite challenge [102]. More recently, a first-in-human phase 1 study demonstrated that the full-length MSP1 vaccine was safe and immunogenic. The MSP1-specific antibodies in the vaccinees activated complement, opsonized merozoites, and triggered respiratory bursts in neutrophils. However, vaccine-induced human antibodies did not show in vitro GIA activity [116,117].

### 4.3. MSP2 (Combination B Vaccine)

MSP2 is a roughly 25 kDa protein that is expressed on the merozoite surface. It is divided into two distinct types, 3D7 and FC27, which are defined based on their central variable regions. The regions surrounding this variable part are highly conserved and categorized into N- and C-terminal regions [118,119]. The MSP2 protein is intrinsically disordered and lacks a known 3D structure, with the exception of a single disulfide bond in the C-terminal region. The N-terminal region tends to form an α-helical structure [120].

A malaria vaccine called Combination B comprises the blood-stage proteins of *P. falciparum*, including K1 strain-type MSP1 (blocks 3 and 4), 3D7-type full-length MSP2, and FC27-type C-terminal ring-infected erythrocyte surface antigen (RESA) (70% of the region). This vaccine formulation was developed with the Montanide ISA720 adjuvant and was tested in Papua New Guinea [85]. A phase 2b study showed that the Combination B vaccine group had lower infection rates for the homologous 3D7-MSP2 parasite, indicating allele-specific protection in naturally exposed populations. In a phase 1 trial of a vaccine containing both 3D7 and FC27 types of MSP2, the induced antibodies showed ADCI activity but required reformulation due to reactogenicity [121]. MSP2 chimeras were also produced, which consisted of the variable region of both the 3D7 and FC27 types and the conserved regions. The immunization of animals with recombinant proteins resulted in a significant antibody response to both types [122].

In further attempts to overcome strain-specificity, researchers have explored targeting epitopes that are located exclusively within conserved regions. Specifically, the C-terminal conserved region of MSP2 was recognized by mouse monoclonal antibodies (mAb). Although these five mAbs bind to overlapping epitopes [123], they exhibit different binding characteristics, and notably, mAbs 4D11 and 9G8 strongly recognized MSP2. Recently, a novel strategy to design a vaccine antigen based on its structure was employed for MSP2, using the crystal structure of the 4D11 variable fragment complexed with its minimal target epitope. This approach involved molecular dynamics simulations and surface plasmon resonance techniques to design multiple constrained peptides that mimic the 4D11-bound epitope structure. These peptides were used to immunize mice, which resulted in the generation of antibodies in all groups. The specificity of antibody responses revealed that a single point mutation can refocus the antibody response to a more favorable epitope [124]. These promising advancements in MSP2-based vaccine design hold great potential to strategize more effective malaria vaccine antigens.

### 4.4. MSP3 and GMZ2

MSP3 is a 48 kDa protein expressed on the merozoite surface and forms a protein complex with MSP1 [114]. The C-terminal region of MSP3 is highly conserved [125], and a long synthetic peptide (MSP3-LSP) derived from this region has been developed as a vaccine antigen for clinical trials [126,127]. In a phase 1b study, Burkinabe children who received the MSP3-LSP vaccine formulated with aluminum hydroxide adjuvant showed significantly fewer clinical malaria episodes compared with the control group, although this trial was initially not designed to measure vaccine efficacy [128].

GMZ2 is an MSP3-based vaccine made of a recombinant protein fusion of MSP3 with a part of the 145 kDa glutamate-rich protein (GLURP) [129,130,131]. Although the function of GLURP remains largely unknown, the GMZ2 vaccine elicited functional antibodies in a phase 1 study [86]. The efficacy of GMZ2 has been evaluated through a multicenter phase 2b trial in children from sub-Saharan countries [86]. The vaccine was immunogenic and well-tolerated, and the vaccine efficacy (VE) was 14% in children who received three doses of the vaccine, as per protocol analysis, adjusted for age and site.

In further attempts to improve efficacy, a recent study conducted on 50 Gabonese adults evaluated the efficacy of two formulations: GMZ2/Alhydrogel and GMZ2/CAF01. The study, using CHMI by intravenously injecting live *P. falciparum* sporozoites (PfSPZ Challenge) to assess vaccine efficacy, showed that the use of the novel adjuvant CAF01 did not significantly affect vaccine efficacy [132]. Despite being immunogenic, MSP3-based vaccines face significant challenges in achieving improved protective efficacy [133].

### 4.5. EBA175

During the invasion process by *P. falciparum*, the 175 kDa merozoite surface receptor EBA175 binds to the sialic acid residues of erythrocyte glycophorin A [53,134]. The binding activity is mainly limited to Region II of EBA175 (RII); therefore, this domain is a potential candidate for a malaria vaccine [55,134]. A phase 1a clinical trial was conducted in malaria-naïve US adults using a vaccine antigen called EBA175 RII-NG, which was expressed without glycosylation in *Pichia pastoris* and was formulated with aluminum phosphate adjuvant. The vaccine was safe, immunogenic, and induced human antibodies showing GIA activity [135]. The vaccine was further assessed for safety and immunogenicity in semi-immune Ghanaian adults in a phase 1b trial. The study showed that the EBA175 RII-NG vaccine was well tolerated, safe, and immunogenic in semi-immune Ghanaian adults [87]. Although recommended for further vaccine development, it is critical to note that different field parasite isolates express varying levels of EBA and Rh proteins, including EBA175, EBA140, EBA181, Rh1, and Rh2, owing to redundant merozoite invasion pathways [136]. This implies that EBA175 alone is inadequate as a vaccine antigen against *P. falciparum,* and a multivalent vaccine that combines other blood-stage proteins should be developed. This has not yet been attempted.

### 4.6. SERA5 (BK-SE36 Vaccine)

The serine repeat antigen 5 (SERA5) of *P. falciparum* is abundantly expressed in schizonts and is secreted into the parasitophorous vacuole to play a vital role in the survival of the parasite. Upon merozoite egress, the SERA5 protein undergoes sequential proteolytic processing to generate five distinct fragments, namely, P47, P73, P50, P6, and P18. This cleavage process involves the initial processing of the 120 kDa precursor protein into P47 and P73 fragments, which are then processed into smaller fragments [137,138,139,140,141]. Epidemiological studies have shown that antibody levels specific to the N-terminal P47 region of SERA5 are negatively correlated with malaria symptoms [142]. Worldwide, the SERA5 sequences of *P. falciparum* are highly conserved and show no evidence of positive selection [143]. African strain variations did not appear to affect the immune response to BK-SE36, with sequence analyses performed on strains collected from clinical trials and follow-up studies in Africa [144]. Immune-target epitopes in the P47 region that are recognized in humans have been identified in N-terminal repetitive sequences [145]. In Uganda, a phase 1b trial of the SE36 vaccine antigen, composed of the P47 region, excluding the serine repeats and formulated with aluminum hydroxide (BK-SE36), demonstrated that the vaccine was safe and immunogenic [146]. A follow-up study provided promising efficacy signals (against clinical malaria) [146] and boosting in vaccinated volunteers because of natural infection [147]. The vaccine was also safe and immunogenic in Burkinabe children, with a higher antibody response in the younger age group [148]. Recently, a novel adjuvant, CpG-ODN(K3), was shown to increase the immune response, particularly in malaria-immune individuals or those with a history of repeated malaria infection. The first-in-human BK-SE36/CpG-ODN(K3) trial in Japan demonstrated that the vaccine was safe and immunogenic [149]. Recently, a phase 1b trial of the BK-SE36/CpG-ODN(K3) vaccine in healthy malaria-exposed Burkinabe adults and children reported that the vaccine was well-tolerated and immunogenic [89]. These findings provide a foundation for additional proof-of-concept investigations aimed at demonstrating the effectiveness of the vaccine.

### 4.7. Rh5

Rh5 is a 45 kDa component of the Rh5/CyRPA/Ripr (RCR) merozoite protein complex [70,138]. Rh5 induces strain-transcending antibodies in animals [72]. Although the levels of these antibodies are low in malaria-exposed individuals [72,74], human anti-Rh5 IgGs have shown growth inhibitory activity in vitro [74,150]. Full-length Rh5 (RH5_FL) vaccination in animals has also induced high growth-inhibitory antibodies against *P. falciparum* lab strains and field-isolated parasites [72,151,152,153]. In a study conducted in *Aotus* monkeys, immunization with 3D7-type Rh5, using ChAd63 for priming and MVA for boosting, protected the animals against challenge infection with heterologous FVO [112].

The first clinical trial of the Rh5 vaccine, based on 3D7-type Rh5, was conducted in healthy adults. The vaccine was administered with the viral vectors ChAd63 and MVA at 8-week intervals and was well tolerated. The Rh5 vaccine elicited antibodies that can effectively inhibit the growth of various parasite strains. The antibodies were capable of identifying both linear and conformational epitopes within Rh5. This is the first evidence where significant Rh5-specific immunity has been demonstrated in humans [154]. In order to assess safety, immunogenicity, and efficacy, a phase 1/2a clinical trial of the RH5.1 (recombinant Rh5 protein) formulated with AS01B adjuvant was conducted in healthy UK adults [91]. The RH5.1/AS01B study demonstrated a significant reduction in the parasite multiplication rate (PMR) following blood-stage CHMI. Thus, this clinical trial was a milestone in the field of asexual blood-stage malaria vaccines. In a subset of vaccine recipients who received an additional booster dose, a stronger inhibition in PMR was observed, which confirmed the findings. Although the vaccine was effective, it only caused a slight delay of 1 to 2 days in the time taken to diagnose the disease [71]. In order to achieve greater protection, research has focused on improved immunogens and formulations that target Rh5 or its invasion complex; for example, a computational approach to design RH5 vaccine antigen variants [155] and structure-guided vaccine antigen development [31,156]. The following clinical trials are currently underway, including a multistage *P. falciparum* malaria vaccine using Matrix-M with RH5.2 VLP and/or R21 (accessed at ClinicalTrials.gov: NCT05790889, NCT05357560, NCT04318002, and NCT05385471).

### 4.8. Ripr

Ripr is another potential antigen that targets the RCR complex. Strain-transcending GIA activities in *P. falciparum* were observed with specific antibodies raised against recombinant Ripr [70]. Rabbit anti-Ripr antibodies showed superior GIA activity to anti-Rh5 antibodies, as reported by Healer et al. [157]. In our own study, we produced a region of Ripr (K279-D995) using the wheat germ cell-free system (WGCFS) [158], and anti-Ripr 3D7-type antibodies significantly inhibited parasite growth against both 3D7 and heterologous FVO strains [159]. Ripr poses a significant challenge as a vaccine target due to its large size of 126 kDa and high cysteine content of 87 residues. In order to address this, we produced 11 truncated fragments of Ripr and generated fragment-specific polyclonal antibodies in animals. Using these antibodies in GIA, we found that a specific Ripr fragment, PfRipr5, spanning C720-D934, was the most potent antigen. It induces the strongest growth-inhibitory antibodies at a level comparable to that of anti-full-length Ripr antibodies [160]. Our findings demonstrate that PfRipr5 is a promising antigen for a novel asexual blood-stage vaccine against *P. falciparum*.

The PfRipr5 recombinant protein was successfully produced on a large scale using the insect High Five cells/baculovirus system for further testing cGMP-compliant methods. The purified PfRipr5 was thermally stable and highly pure and demonstrated a high binding capacity to anti-PfRipr5 functional mAb [161]. Purified PfRipr5 was tested in combination with licensed adjuvants, such as Alhydrogel, GLA-SE, and CAF01, and it showed acceptable compatibility and produced comparable levels of anti-PfRipr5 antibodies in rabbits. The highest GIA activity was observed when PfRipr5 was formulated with the CAF01 adjuvant [93]. Further preclinical and clinical studies are planned to develop a PfRipr5-based vaccine. In addition, the fine structure of Ripr in the RCR complex was recently reported [31]. It may be useful for the future Ripr-based vaccine design.

### 4.9. CyRPA

CyRPA is another essential merozoite antigen and is a member of the RCR complex. Anti-CyRPA antibodies have strain-transcending GIA activities and synergistic effects with anti-Rh5 [28]. Naturally acquired antibodies against CyRPA prevent parasite invasion of erythrocytes and reduce malaria episodes [162,163]. The virosome-based CyRPA vaccine has been shown to elicit antibodies that are protective both in vitro and in vivo [164]. Somanathan et al. [165] successfully developed a large-scale production process for CyRPA with the refolding of tag-free CyRPA antigen in *E. coli*. Alhydrogel-formulated tag-free CyRPA induced a high antibody response exhibiting strain-transcending GIA activity. Another approach for establishing scalable, cost-effective, robust, and high-yield CyRPA production processes was successfully achieved using High Five insect cells [166]. When formulated in lipid-based virosome nanoparticles, rabbit anti-CyRPA antibodies showed potent GIA activities [166]. Recently, the recombinant CyRPA antigen, produced in mammalian HEK293 cells and formulated with GLA-SE adjuvant, was also immunogenic in animals and induced anti-CyRPA antibodies with potent GIA activities [95]. Further development efforts are required.

### 4.10. P27A

A genome-wide approach has been used to identify new vaccine-candidate antigens that are potential targets for growth-inhibitory antibodies [167]. One of the targets identified through this approach was a protein fragment known as P27A (a part of TEX1) [75,168]. The P27A fragment is intrinsically unstructured [167]. Human antibodies that target P27A are capable of inhibiting parasite invasion in vitro, and the immune response to P27A has been linked to protection against malaria [167,168]. Clinical trials were conducted to evaluate the safety and efficacy of a P27A vaccine, with adjuvants Alhydrogel or GLA-SE, in both malaria-naïve individuals from Switzerland and healthy adults from Tanzania [98]. The vaccine was safe, induced high levels of antibodies (especially when used with GLA-SE), and exhibited effective growth-inhibitory activities [98].

### 4.11. PfSEA-1

PfSEA-1 is a 244 kDa protein expressed in schizont-infected erythrocytes that plays a crucial role in blood-stage parasite growth. In malaria-endemic areas, the development of antibodies against PfSEA-1A (aa 801–1023) was observed [76]. In vitro studies have shown that these antibodies against PfSEA-1A inhibit parasite egress from schizont-infected erythrocytes. Studies conducted on Tanzanian children and Kenyan adults have revealed that individuals with antibodies to recombinant PfSEA-1A experience a significant reduction in parasitemia and the incidence of severe malaria [76]. Recently, maternally derived antibodies against PfSEA-1A in cord blood reduced severe malaria in infants. These observations were also replicated using a rodent malaria model [99]. The results suggest that the vaccination of pregnant women with PfSEA-1 may provide their offspring with a survival advantage [99]. Despite these promising findings, further vaccine development efforts on PfSEA-1 have not been reported.

### 4.12. PfGARP

Two separate groups discovered PfGARP. Almukadi et al. [169] used human erythrocytes as bait to screen the phage cDNA libraries of *P. falciparum*. They found a clone of PfGARP (aa 356–552; PfGARP-L) and a smaller PfGARP fragment (aa 392–437; PfGARP-S). PfGARP is a novel erythrocyte-binding protein that uses band 3 as its erythrocyte receptor. Raj and colleagues [77] performed differential biopanning using a T7 bacteriophage cDNA library of blood-stage *P. falciparum* 3D7. They identified that PfGARP is a protective antigen recognized by antibodies in children with immunity to *P. falciparum*. PfGARP is an 80 kDa protein that contains a *Plasmodium* export element (PEXEL) motif, enabling its translocation into infected erythrocytes and expression on the infected erythrocyte surface. Non-human primates vaccinated with PfGARP-based mRNA or protein vaccines showed partial protection against *P. falciparum* challenge. Longitudinal studies have shown that Tanzanian children and Kenyan adults who lack anti-PfGARP antibodies are at a greater risk of contracting severe malaria [77]. Sequence analysis confirmed that PfGARP is highly conserved, suggesting that a PfGARP-based vaccine may elicit strain-transcending immunity [170]. Similar to PfSEA-1, further vaccine development efforts have not been reported.

### 4.13. Others

In this review article, we did not describe two types of asexual blood-stage vaccines of *P. falciparum* that are under clinical development, namely, placental malaria vaccines [171,172,173], including the structural design of a VAR2CSA vaccine antigen [174] and whole blood-stage parasite vaccines [32,175,176,177,178,179].

## 5. Conclusions

Recently, decades of research on malaria vaccines have resulted in significant advances, culminating in the recommendation of the RTS,S/AS01 and R21 vaccines by the WHO. The progress suggests that a promising strategy to interrupt the parasite lifecycle at multiple stages could involve using a combination of pre-erythrocytic-stage vaccines (such as RTS,S/AS01 or R21) that are partially effective, along with asexual blood-stage vaccines. This combination can theoretically prevent parasite infection and the development of a blood-stage infection in the human host. Currently, the most advanced asexual blood-stage malaria vaccines are AMA1-based FMP2.1/AS02A [80] and MSP3-based GMZ2 [86]. They were tested in phase 2b trials; however, the vaccines had no significant clinical efficacy.

The high polymorphism levels in *P. falciparum* vaccine antigens are a major reason why the vaccine trials of leading asexual blood-stage targets were unsuccessful [80,81,83,85]. These polymorphisms often result in strain-specific immunity that hampers vaccine efficacy in clinical trials. Various studies that attempted to address the effects of allele-specific protective efficacy by combining several diversity-covering variants were met with a number of challenges [104]. Therefore, emerging evidence suggests that conserved antigens across multiple strains could be a more straightforward approach to attain high protective efficacy in the field [112]. Until recently, only one vaccine candidate has been considered to be such a conserved blood-stage vaccine candidate, namely, *P. falciparum* reticulocyte-binding homolog 5, Rh5, and it showed promising vaccine efficacy in a phase 1/2a trials [91]. As described above, a number of novel blood-stage vaccine candidates have been reported after complete sequence information for multiple parasite genomes has become available; however, unfortunately, to date, no additional asexual blood-stage vaccines have proceeded to phase 2b clinical trials. So far, the most advanced vaccine candidates are Rh5 and BK-SE36 vaccines, but proof-of-concept trials have not yet been reported. Thus, further genome-wide discovery efforts for a novel, highly conserved asexual blood-stage vaccine candidates are urgently needed.

The history of malaria vaccine development summarized in this review dissects the approaches to antigen discovery, protein expression technologies, the use of adjuvants, alternative delivery platforms, and the key lessons learned in the iterative phases of vaccine development. CHMI is now a popular tool in the early clinical phase that can significantly accelerate the vaccine developmental timelines. Numerous vaccine clinical trial sites have been established in malaria-endemic countries. These dramatic advances emphasize to the research community that a pipeline is available for the successful development of second-generation malaria vaccines. In order to achieve success in developing a vaccine for asexual blood-stage malaria, it is crucial to prioritize genome-wide identification of novel vaccine antigens.

## Figures and Tables

**Figure 1 biomolecules-14-00100-f001:**
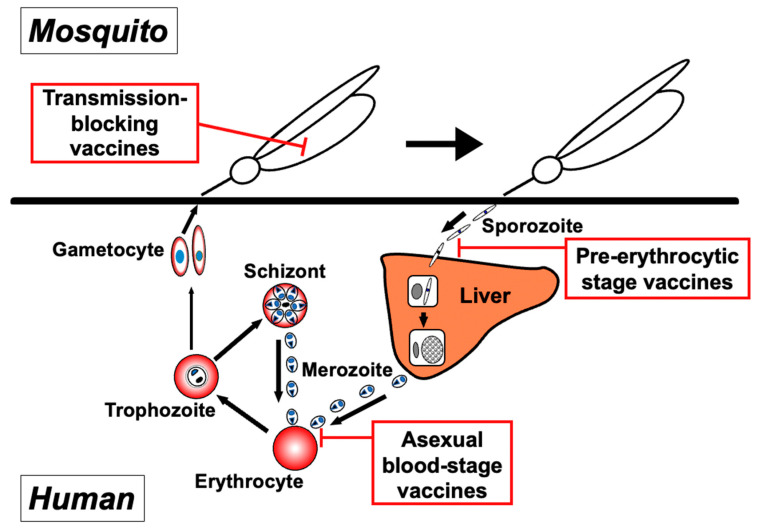
Targets of three approaches to malaria vaccines within the parasite lifecycle. Malaria vaccine development can be categorized into three groups, with each targeting distinct parasite developmental stages: pre-erythrocytic-stage vaccines, asexual blood-stage vaccines, and transmission-blocking vaccines.

**Figure 2 biomolecules-14-00100-f002:**
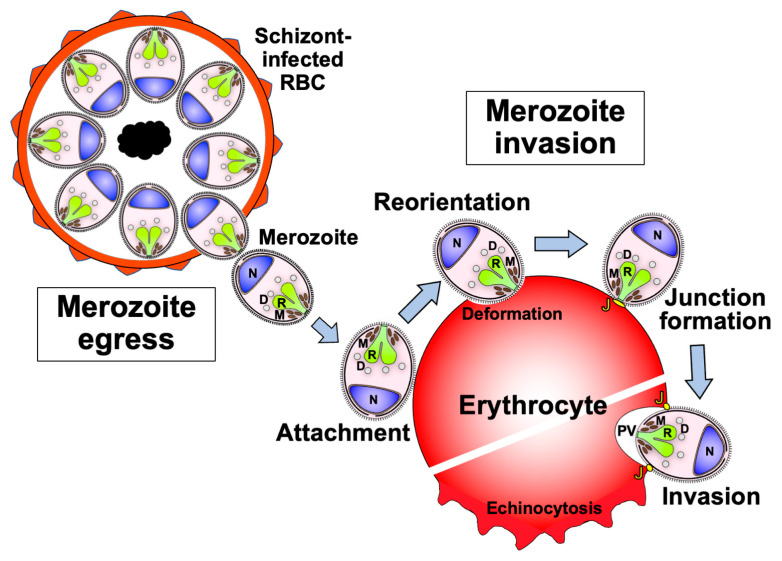
Schematic representation of the sequential events of the merozoite from egress to erythrocyte invasion. Soon after the merozoite egresses from the schizont, it attaches to the surface of a new erythrocyte. Then, the merozoite reorients its apical end toward the erythrocyte. After reorientation, the merozoite forms a tight junction between its surface and the erythrocyte surface. Finally, the merozoite starts to invade the erythrocyte via a moving junction. Once invasion begins, the infected erythrocyte plasma membrane deforms and exhibits echinocytosis until the completion of invasion. In addition to the merozoite surface proteins, the secretory organelles of the merozoite are known to contain the necessary molecules for its invasion. M: microneme organelle; R: rhoptry organelle; D: dense granule organelle; J: junction between the merozoite surface and erythrocyte plasma membrane; PV: parasitophorous vacuole that surrounds the invading merozoite; N: nucleus.

**Table 1 biomolecules-14-00100-t001:** Clinical development of the asexual blood-stage malaria vaccine candidates of *P. falciparum*.

Antigen ^a^	Vaccine ^b^	Most Advanced Outcome ^c^	Structure ^d^
AMA1	FMP2.1/AS02A	P2b: Strain-specific [80,81]	Solved [82]
MSP1	FMP1/AS02A	P2b: Ineffective [83]	Solved [84]
MSP2	Combination B	P2b: Strain-specific [85]	ND
MSP3	GMZ2	P2b: Ineffective [86]	ND
EBA175	EBA175 RII-NG	P1b: Safe, immunogenic [87]	Solved [88]
SERA5	BK-SE36/CpG	P1b: Safe, immunogenic [89]	Solved [90]
Rh5	RH5.1/AS01B	P2a: Reduction of PMR [91]	Solved [31,92]
Ripr	PfRipr5/CAF01	Pre-clinical [93]	Solved [31,94]
CyRPA	CyRPA/GLA-SE	Pre-clinical [95]	Solved [96,97]
P27A	P27A/GLA-SE	P1b: Safe, immunogenic [98]	ND
PfSEA-1	PfSEA-1/TiterMax	Experimental [99]	ND
PfGARP	PfGARP/mRNA	Pre-clinical [77]	ND

^a^ List of asexual blood-stage vaccine antigens; ^b^ List of formulated vaccines; ^c^ Most advanced outcomes; ^d^ Tertiary structure data. P1: phase 1 clinical trial, etc.; Strain-specific, parasite strain-specific protection; PMR: parasite multiplication rate in vivo; Experimental: not advanced to pre-clinical development; ND: not determined.

## Data Availability

Not applicable.

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
