# Peer review of "The Need for Novel Asexual Blood-Stage Malaria Vaccine Candidates for Plasmodium falciparum"

_biomolecules, 2024, doi:10.3390/biom14010100_

Round 1
Reviewer 1 Report
Comments and Suggestions for Authors
This review is very comprehensive and covers the latest advances in malaria research in detailed and robust manner including latest developments in both liver and blood-stage malaria. t
It is advised to add the date of vaccine of approval by WHO for R21-Matrix vaccine.
There are a few essential references which would need to be added to different protein vaccine targets, where the author cited reviews instead of original research, which is not a good practice.
Please see some suggestion below of key ones:
RH5 target:
It needs to be mentioned that a vaccine is in the pipeline for R21-Matrix RH5.2 as a follow up to the current R21-Matrix one
ps://doi.org/10.1073/pnas.1616903114
https://www.cell.com/med/pdfExtended/S2666-6340(21)00116-1
https://pubmed.ncbi.nlm.nih.gov/31204103/
https://www.biorxiv.org/content/10.1101/2022.09.23.509221v1
PfEMP1 target:
https://www.ncbi.nlm.nih.gov/pmc/articles/PMC6972667/
VAR2CSA target:
many references are reviews, again it is better to avoid a chain of references to reviews in a review.
https://doi.org/10.1038/s41467-021-23254-1
Also, given the central role of structural biology for malaria vaccine design I strongly recommend to showcase in a Figure a gallery of the structures solved to date of the above proteins by x-ray crystallography and cryo-EM.
Author Response
Please see the attachment, which includes responses to both Reviewers 1 and 2.

Reviewer 2 Report
Comments and Suggestions for Authors
The review by Takashima et al entitled ‘The need for novel blood-stage malaria vaccine candidates for Plasmodium falciparum’ providers a comprehensive scientific literature overview of malaria vaccines directed against asexual blood stages of the P. falciparum parasite.
I am proposing a few minor changes to be considered by the authors.
1) The title refers to blood stage malaria vaccines candidates, which I think could be improved be referring to asexual blood stages, as this will exclude gametocytes as a sexual blood stage.
2) I think that a table (or figure) listing all malaria vaccine candidates described in the review will be informative for the reader, and suggest to update the review accordingly.
3) The rationale for describing the vaccine candidates in the context of a pre- and post-genomics era is a little unclear to me, given that paragraph 4 ‘Asexual blood-stage malaria vaccine candidate discovery in the post-genomic era’ is relatively short.
4) Identifying malaria vaccine candidates against asexual blood stages is complicated by immune evasion mechanisms of the parasite. A dedicated paragraph about immune evasion by the parasite would provide the need for developing novel vaccine candidates. Therefore, I hope the authors could consider adding a paragraph about this topic.
Comments on the Quality of English LanguageThe review is written in appropriate English
Author Response

(The authors gave the same response as above.)
